A happier and less sinister past, a more hedonistic and less fatalistic present and a more structured future: time perspective and well-being

Sailer Uta 1 2
Rosenberg Patricia 2
Nima Ali Al 1 2
Gamble Amelie 1
Gärling Tommy 1
Archer Trevor 1 2
Garcia Danilo 2 3 danilo.garcia@neuro.gu.se,danilo.garcia@euromail.se
1 Department of Psychology, University of Gothenburg , Göteborg , Sweden
2 Network for Empowerment and Well-Being , Sweden
3 Centre for Ethics, Law and Mental Health (CELAM), University of Gothenburg , Göteborg , Sweden
Cloninger C. Robert
Electronic publication date: 2014 Mar 11
Publication date: 2014
Volume: 2
Electronic Location ID: e303
Received 2013 Dec 17; Accepted 2014 Feb 12
Copyright: © 2014 Sailer et al.
Copyright year: 2014
Copyright holder: Sailer et al.
License: This is an open access article distributed under the terms of the Creative Commons Attribution License, which permits unrestricted use, distribution, and reproduction in any medium, provided the original author and source are credited.
License URL: https://creativecommons.org/licenses/by/3.0/

Keywords: Positive affect, Psychological well-being, Subjective well-being, Temporal life satisfaction, Negative affect, Time perspective

Funding: The Swedish National Centre for Research in Sports P2012-0097 AFA Insurance grant to Danilo Garcia The development of this article was supported by The Swedish National Centre for Research in Sports (Grant nr. P2012-0097) and AFA Insurance. The funders had no role in study design, data collection and analysis, decision to publish, or preparation of the manuscript.

==============================
Background. Previous studies have established a link between how people relate to their past, present, and future (i.e., time perspective) and subjective well-being (i.e., life satisfaction, positive and negative affect). Time perspective comprises five dimensions: Past Positive, Past Negative, Present Hedonistic, Present Fatalistic, and Future. Life satisfaction can also be evaluated in relation to different time frames. Moreover, approach related positive affect is associated to a different concept of well-being labeled psychological well-being. In the present study we extend previous findings by investigating the effect of time perspective on the time frame of evaluations of life satisfaction (past, present, future) and by investigating the relationship between time perspective and psychological well-being.

Method. Questionnaires on time perspective (Zimbardo’s Time Perspective Inventory), temporal life satisfaction (Temporal Satisfaction with Life Scale), affect (Positive Affect and Negative Affect Schedule), and psychological well-being (Scales of Psychological Well-Being—short version) were answered by 453 individuals. Two different structural equation models were tested, one of the relationship between time perspective and temporal life satisfaction, and the other of the relationship between time perspective, affect and psychological well-being.

Results. Time perspective affected life satisfaction depending on the time scale on which it was evaluated—memory of a negative past influenced life satisfaction in all time frames, and a positive view of the past influenced both past and future life satisfaction. Moreover, less rumination about past negative events (i.e., low score on Past Negative), the tendency to take risks in the present to achieve happy feelings and/or avoid boredom (i.e., high scores on Present Hedonistic), and a less hopeless and pessimistic view about the present (low scores on Present Fatalistic) were associated with higher levels of psychological well-being and positive affect. These same time perspective dimensions were associated with lower levels of negative affect. The Future time perspective dimension (i.e., approaching life with self-control, punctuality, and planning for the future) was associated with both psychological well-being and positive affect.

Conclusions. High levels of both subjective and psychological well-being are related to a happier and a less sinister past, a more hedonistic and less fatalistic present, as well as to a more structured future.

Introduction

Humans anchor events and experiences in time to make sense of them. For example, they may use previous instances of an event to predict its occurrence in the future. Zimbardo and colleagues (e.g., Zimbardo, 2008; Zimbardo & Boyd, 1999) have developed a model of how people organize and apprehend time. Their time-perspective model consists of five dimensions: (i) Past Positive reflecting a sentimental and positive view of the past; (ii) Past Negative which reflects a pessimistic attitude toward the past; (iii) Present Hedonistic reflecting the desire of experiencing pleasure with slight concern for future consequences; (iv) Present Fatalistic which reflects a lack of hope and control for the future, and; (v) Future which reflects the ability to find reward in achieving specific long-term goals. Time perspective is expected to influence attitudes, behaviour and goals.

Empirically it has been shown that time perspective predicts the reported use of alcohol, drug, and tobacco (Keough, Zimbardo & Boyd, 1999), risky driving (Zimbardo, Keough & Boyd, 1997), indecision and avoidant procrastinations (Díaz-Morales, Ferrari & Cohen, 2008), environmental engagement (Milfont, Wilson & Diniz, 2012), the choice of food and of partner, educational achievement, and the distinctness of future goals (Zimbardo & Boyd, 1999). Time perspective is also related to mood. As an example, negative views of the past have been reported to be associated with depression, whereas positive views of the past have been reported to be associated with happiness (Zimbardo & Boyd, 1999; see also Stolarski et al., 2013). The influence of a past-oriented time perspective may be explained by episodic-memory retrieval that influences how people imagine and simulate future events (Schacter & Addis, 2007a; Schacter & Addis, 2007b; Schacter & Addis, 2007c). This view is supported by episodic-memory retrieval that influences how people imagine and simulate future events (Addis, Wong & Schacter, 2007; Szpunar, Watson & McDermott, 2007).

Subjective well-being or happiness is defined as a person’s cognitive and affective evaluations of her/his life (Diener, Lucas & Oishi, 2002). It is influenced both by judgments of life satisfaction and fulfillment as well as the balance between the frequency and intensity (or duration) of positive emotions and moods and the frequency and intensity (or duration) of negative emotions and moods. Life-satisfaction judgments are believed to be based on memories of current and previous life experiences that cause evaluative and emotional reactions (Kim-Prieto et al., 2005).

Previous research suggests that subjective well-being may be critically related to time perspective. Thus, Zimbardo & Boyd (1999) proposed that a ‘balanced time perspective’ is important for optimal functioning. This implies an ability to simultaneously evaluate past, present, and future time perspectives in a flexible manner. In accordance with Zimbardo & Boyd’s (1999) assumption, individuals with such a balanced time perspective report greater happiness (Drake et al., 2008). Furthermore, subjective well-being is higher for individuals with a positive view of the present and past than for individuals with a negative view of the past (Zhang & Howell, 2011). Whereas these studies demonstrate an effect of time perspective on subjective well-being, it is not known whether this effect also depends on the time frame (past, present and future) in which individuals evaluate their well-being. Moreover, life-satisfaction judgments may be interrelated, that is, that past life satisfaction is related to present and future life satisfaction (Kim-Prieto et al., 2005; Garcia, Rosenberg & Siddiqui, 2011).

Arguing that well-being goes beyond happiness and satisfaction with life, Ryff (1989) proposed the concept of psychological well-being—while subjective well-being focuses on the pursuit of happiness and a pleasant life, psychological well-being targets the fulfillment of human potential and a meaningful life even in the face of challenges and adversities (Ryan & Deci, 2001). Psychological well-being is conceptualized as containing six dimensions: self-acceptance (i.e., the ability to accept all parts of the self), personal growth (i.e., seeing life as an opportunity to develop), purpose in life (i.e., having a sense of meaning in life), environmental mastery (i.e., the sense of having control in one’s life), autonomy (i.e., the sense of self-directedness), and positive relations with others (i.e., the ability to establish and keep warm, close, and trustful relations with others). Thus, psychological well-being has a wider focus on the achievement of full psychological potential and functioning. This concept has been found to be strongly related to subjective well-being (Compton et al., 1996). Urry and colleagues (2004), for example, investigated correlations between individual differences in baseline prefrontal activation and psychological well-being. They found that affect, especially approach-related positive affect (e.g., feeling “interested” or “strong”), is related to psychological well-being (engaging with goal-directed stimuli). Hence, psychological well-being, as well as positive affect, seems to have an orientation towards the future. The concept of approach-related positive affect is, for instance, related to Gray’s (1981) Behavioural Activation System or sensitivity to reward as well as approach motivation, while negative affect is related to the Behavioural Inhibition System or sensitivity to punishment as well as avoidance motivation. Moreover, recent research (Garcia, 2013) has found that individuals with high levels of psychological well-being recall more positive than negative memories from the past. How individuals relate to the past may also be related to their own self-acceptance, personal growth, purpose in life, environmental mastery, autonomy, and ability to create positive relations with others.

The present study

Only one study has explored Zimbardo’s time perspective model and temporal judgments of life satisfaction (Boniwell et al., 2010). This study used the instrument developed to measure temporal variations of life satisfaction (e.g., Temporal Satisfaction With Life Scale; Pavot, Diener & Suh, 1998) that will also be used in the present study. Moreover, the present study aims to close another knowledge gap in the literature, namely how different time perspective dimensions are related to psychological well-being. Ryff’s multidimensional model of psychological well-being is a point of departure in the present study which, although related, is conceptually different to subjective well-being (Kjell et al., 2013).

As detailed in the introduction, temporal satisfaction with life seems to be an interrelated construct, that is, past life satisfaction is related to present and future life satisfaction and vice versa. Thus, the first aim of the present study is to investigate the relationship between the dimensions of time perspective and temporal life satisfaction taking into account the inter-relation between past, present, and future life satisfaction judgments. The second aim is to investigate the relationship between time perspective dimensions and psychological well-being and affect by taking into account the close relationship between psychological well-being and affect, especially positive affect.

Method

Ethics statement

Data collection conformed to the Declaration of Helsinki and the Ethics Committee of the University of Gothenburg approved the research protocol. Verbal informed consent was obtained from all the study participants as agreed by the review board.

Participants and procedure

The data were collected among 400 undergraduate students at the University in the West of Sweden (from which we obtained 324 valid responses corresponding to a 81% response rate) and at a training facility, a gym complex for weight and aerobic training, in the South of Sweden (N = 158 from which we obtained 129 valid responses corresponding to a 82% response rate). The data collected at the training facility was supported by a grant from The Swedish National Centre for Research in Sports (Grant nr. P2012-0097). A total of 453 individuals participated in the study (148 males and 300 females, 5 who failed to report their gender, mean age 29.74 years SD = 12.86 years). All participants were informed that their participation was voluntary and anonymous. They were presented with a battery of instruments used to collect the relevant measures in the following order: background, time perspective, temporal satisfaction with life, psychological well-being, and affect. Although randomizing the order in which the instruments are presented to participants is suggested to ensure no order effects (Lavrakas, 2008) this was not practically possible to do in both samples. While the university group answered a paper and pencil version of the instruments, the gym group answered an online version by receiving a link to their email addresses and were asked to answer the questionnaires in the tranquility of their homes. The gym group also differed in that those who participated received a cinema ticket for their collaboration, whereas no compensation was offered to the undergraduate group.

Measures

Background and health questionnaire (Karlsson & Archer, 2007). This instrument was applied to collect background data providing health and health-related information about each participant. The questionnaire consists of items pertaining to age, gender, education, sleeping problems, propensity to perform regular physical exercise, and use of psychotropic drugs.

Time perspective. The Zimbardo Time Perspective Inventory (Zimbardo & Boyd, 1999) consists of 56 items that measure the following five time dimensions: Past Positive (e.g., “It gives me pleasure to think about my past”), Past Negative (e.g., “I think about the good things that I have missed out on in my life”), Present Hedonistic (e.g., “Taking risks keeps my life from becoming boring”), Present Fatalistic (e.g., “Fate determines much in my life”), and Future (e.g., “I believe that a person’s day should be planned ahead each morning”). The Swedish version has been used and validated in previous studies (Carelli, Wiberg & Wiberg, 2011; Wiberg et al., 2012) and its psychometric properties validated in many different languages, such as Portuguese (Milfont et al., 2008), Lithuanian (Liniauskaite & Kairys, 2009), and Spanish (Díaz-Morales, 2006). The Cronbach’s αs in the present study were: .81 for Past Positive, .87 for Past Negative, .81 for Present Hedonistic, .68 for Present Fatalistic, and .77 for Future.

Temporal life satisfaction. The Temporal Satisfaction With Life Scale (Pavot, Diener & Suh, 1998) comprises 15-items rated on a 7-point Likert scale (1 = strongly disagree, 7 = strongly agree) organized in three subscales assessing past (e.g., If I had my past to live over, I would change nothing), present (e.g., I would change nothing about my current life), and future life satisfaction (e.g., There will be nothing that I will want to change about my future). The Cronbach’s α in the present study were: .86 for the past subscale, .93 for the present subscale, and .88 for the future subscale.

Affect. The Positive Affect and Negative Affect Schedule (Watson, Clark & Tellegen, 1988) assesses the affective component of subjective well-being by requiring participants to rate on 5-point adjective scales to what extent (1 = very slightly, 5 = extremely) during the last few weeks they experienced 10 positive and 10 negative affect. The positive affect scale includes adjectives such as strong, proud, and interested; and the negative affect scale includes adjectives such as afraid, ashamed, and nervous. The Swedish version has been used in previous studies (e.g., Garcia & Erlandsson, 2011; Nima, Archer & Garcia, 2012; Nima, Archer & Garcia, 2013; Nima et al., 2013; Schütz, Archer & Garcia, 2013; Schütz, Garcia & Archer, 2014). The Cronbach’s αs in the present study were for positive affect .88 and for negative affect .84.

Psychological well-being. The Scales of Psychological Well-Being-short version (Clarke et al., 2001) comprises 18 items including 3 items for each of the six dimensions. These dimensions are: self-acceptance (e.g., “I like most aspects of my personality”), personal growth (e.g., “For me, life has been a continuous process of learning, changing, and growth”), purpose in life (“Some people wander aimlessly through life, but I am not one of them”), environmental mastery (e.g., “ I am quite good at managing the responsibilities of my daily life”), autonomy (e.g., “I have confidence in my own opinions, even if they are contrary to the general consensus”), and positive relations with others (e.g., “People would describe me as a giving person, willing to share my time with others”). The Swedish version has been used in previous studies (e.g., Garcia, 2011; Garcia & Siddiqui, 2009). Since the subscales have been found to have low reliability, the total psychological well-being score (i.e., the sum of the 18 items) is recommended as a better and more reliable measure (Garcia & Siddiqui, 2009). A Cronbach’s α of .83 was in the present study obtained for the total psychological well-being score.

Statistical treatment

In order to determine whether both samples could be pooled, we first conducted an independent t-test between the two groups (undergraduate and gym participants) using the background variables (i.e., education, sleeping problems, exercise frequency, and use of psychotropic drugs) as the dependent variables. We assumed that non-significant differences between the samples in most of these variables would justify pooling them. The results showed that the groups only differed on exercise frequency (t = 4.65, df = 451, p < .001), the gym group reporting exercising more frequently (M = 4.06) than the undergraduate group (M = 3.59). Since there were no differences in the other variables, all subsequent analyses were conducted on the whole sample (N = 453).

We used the Expectation-Maximization Algorithm (EM-Algorithm) to input missing values. Little’s Chi-Square test for Missing Completely at Random showed a χ2 = 67.25 (df = 53, p = .09) for men and χ2 = 77.65 (df = 72, p = .31) for women. Thus, the EM-Algorithm was suitable to use. Table 1 shows the Pearson’s correlations inter-correlations between all variables. There are significant correlations between many of the variables, which potentially introduces multicollinearity. Thus, the data were analysed using Structural Equation Modeling (SEM) in order to control for error measurement and collinearity among variables. Based on our research questions we conducted two SEM models, one of the relationship between time perspective and temporal satisfaction (Satisfaction Model) and the other one of the relationship between time perspective and both affect and psychological well-being (PANA and PWB model). Kurtosis and skewness values for all variables in the first (highest kurtosis value = .77; highest skewness value = −.65) and in the second SEM model (highest kurtosis value .77; highest skewness value .75) were within acceptable ranges.

Table 1 Correlations between time perspective, subjective and psychological well-being variables.

	1	2	3	4	5	6	7	8	9	10	11	
Past Positive (1)	-											
Past Negative (2)	−.39**	-										
Present Hedonistic (3)	.13**	.05	-									
Present Fatalistic (4)	−.07	.37**	.24**	-								
Future (5)	.01	.03	−.22**	−.15**	-							
Past Satisfaction with Life (6)	.58**	−.63**	.07	−.17**	−.03	-						
Present Satisfaction with Life (7)	.30**	−.52**	.12*	−.20**	−.02	.45**	-					
Future Satisfaction with Life (8)	.31**	−.36**	.17**	−.08	.05	.39**	.53**	-				
Positive Affect (9)	.23**	−.32**	.17**	−.30**	.19**	.24**	.53**	.40**	-			
Negative Affect (10)	−.29**	.58**	−.07	.27**	.04	−.43**	−.49**	−.39**	−.29**	-		
Psychological Well-Being (11)	.41**	−.58**	.14**	−.38**	.07	.49**	.59**	.47**	.53**	−.60**	-	
Notes.

* p < .05.

** p < .01.

The chi-square value was significant for the Satisfaction Model (Chi2 = 3.96, df = 1, p = .047). However, it is known that the chi square statistic is heavily influenced by sample size (Kline, 2010) such that larger samples have a higher likelihood of being significant. Since other measures of fit suggested a good model fit (see below), we considered that the Satisfaction Model was acceptable.

The chi-square value for the PANA and PWB Model was strongly significant (Chi2 = 126.04 df = 3, p < .001) suggesting that the model did not fit the data to an acceptable degree. After modifying the model by adding covariances between errors, the chi-square value of the PANA and PWB Model was found to be acceptable (Chi2 = 3.85, df = 1, p = .05). Corroborating evidence is provided by the Root Mean Square Error of Approximation fit statistic that was below 0.10 (Satisfaction Model: .08 and PANA and PWB Model: .08) indicating a reasonable fit (Browne & Cudeck, 1993). The incremental fit index (Bollen, 1989) indicated that the model fit was acceptable (Satisfaction Model: 1.00 and PANA and PWB Model: 1.00). Finally, the Bentler–Bonett Normed Fit Indices (Bentler & Bonnet, 1980) also indicated that the model fit was acceptable (Satisfaction Model: 1.00 and PANA and PWB Model: 1.00).

Results

The first SEM model was theoretically based on the relationship between the dimensions of temporal life satisfaction (past, present, and future; see Kim-Prieto et al., 2005) and the independency of this construct of subjective well-being (Diener, Lucas & Oishi, 2002). This first model (Satisfaction Model), was conducted to investigate which time perspective dimensions influence temporal life satisfaction using some modifications (i.e., regression weights from past life satisfaction to present life satisfaction and from present life satisfaction to future life satisfaction) to obtain a better fitting for this model. The second model was based on the relation between affectivity measured by the Positive Affect and Negative Affect Schedule and psychological well-being (Urry et al., 2004). This model (PANA and PWB Model) was conducted to investigate which perspective dimensions influence affect and psychological well-being. The possible covariances for the five time perspective dimensions are taken into account in both models.

The results of the Satisfaction Model showed that the Past Positive (β = .38, p < .001) time perspective dimension predicted past life satisfaction, while the Past Negative dimension significantly counterpredicted past life satisfaction (β = −.49, p = < .001). Present life satisfaction was predicted by the Present Hedonistic (β = .14, p < .001) and counterpredicted by the Past Negative (β = −.38, p < .001) time perspective dimensions. Present life satisfaction was also predicted by past life satisfaction (β = .17, p = .002). Future life satisfaction was predicted by the Past Positive (β = .12, p = .005), the Present Hedonistic (β = .11, p = .007), and the Future (β = .09, p = .03) time perspective dimensions. Present life satisfaction was also involved in the prediction of future life satisfaction (β = .43, p < .001), while the Past Negative dimension counterpredicted future life satisfaction (β = −.12, p = .020). The whole model yielded a R2 = .31 for past life satisfaction, .33 for present life satisfaction, and .33 for future life satisfaction. See Fig. 1 and Table 2 for the details.

Figure 1 Structural equation model of time perspective and temporal life satisfaction (i.e., the Satisfaction Model).

Showing all inter-correlations for the five time perspective dimensions, all paths from the time perspective dimensions to the three temporal life satisfaction constructs, and their standardized parameter estimates. Chi-square = 3.96, df = 1, p = .047; comparative fit index = 1.00; incremental fit index = 1.00; normed fit index = 1.00; root mean square error of approximation = 0.08.

Table 2 Standardized and unstandardized coefficients for both models in the study.

Time perspective dimension	Well-being measure	R 2	β	B	SE	p	
Satisfaction Model	
Past Positive			.38	.82	.08	<.001	
Past Negative			−.49	−.91	.07	<.001	
Present Hedonistic	Past Life Satisfaction	.53	.04	.11	.10	.28	
Present Fatalistic			.03	.82	.10	.43	
Future			−.01	−.03	.10	.78	
Past Positive			.03	.06	.11	.56	
Past Negative			−.38	−.73	.10	<.001	
Present Hedonistic	Present Life Satisfaction	.31	.14	.43	.13	<.001	
Present Fatalistic			−.06	−.17	.13	.19	
Future			.02	.06	.12	.65	
Past Positive			.12	.22	.08	.005	
Past Negative			−.12	−.18	.08	.02	
Present Hedonistic	Future Life Satisfaction	.33	.11	.28	.10	.007	
Present Fatalistic			.04	.10	.03	.31	
Future			.09	.22	.10	.03	
							
PANA and PWB Model	
Past Positive			.08	.08	.04	.06	
Past Negative			−.22	−.17	.04	<.001	
Present Hedonistic	Positive Affect	.25	.28	.36	.06	<.001	
Present Fatalistic			−.24	−.30	.06	<.001	
Future			.23	.29	.05	<.001	
Past Positive			−.07	−.06	.04	.11	
Past Negative			.53	.39	.03	<.001	
Present Hedonistic	Negative Affect	.36	−.10	−.12	.05	.01	
Present Fatalistic			.09	.10	.05	.03	
Future			.01	.02	.05	.73	
Past Positive			.20	.17	.03	<.001	
Past Negative			−.43	−.31	.03	<.001	
Present Hedonistic	Psychological Well-Being	.46	.21	.25	.04	<.001	
Present Fatalistic			−.24	−.27	.04	<.001	
Future			.09	.10	.04	.01	
Notes.

Significant relations in bold type.

The results of the PANA and PWB Model showed that positive affect was predicted by the Present Hedonistic (β = .28, p < .001), and the Future (β = .23, p < .001) time perspective dimensions, while the Past Negative (β = −.22, p < .001) and the Present Fatalistic (β = −.24, p < .001) dimensions counterpredicted positive affect. Negative affect was predicted by the Past Negative (β = .53, p < .001) and the Present Fatalistic (β = .09, p = .03) time perspective dimensions, while the Present Hedonistic (β = −.10, p = .01) dimension counterpredicted negative affect. The whole model yielded a R2 = .25 for positive affect and .36 for negative affect. See Fig. 2 and Table 2 for the details. The results of this model also showed that psychological well-being was predicted by all five time perspective dimensions. Specifically, the Past Positive (β = .20, p < .001), the Present Hedonistic (β = .21, p < .001), and the Future (β = .09, p = .010) time perspective dimensions, while the Past Negative (β = −.43, p < .001) and the Present Fatalistic (β = −.24, p < .001) dimensions counterpredicted psychological well-being. The whole model showed an R2 = .46 for psychological well-being. See Fig. 2 and Table 2 for the details.

Figure 2 Structural equation model of time perspective, psychological well-being and affect (i.e., the PWB and PANA Model).

Showing all inter-correlations for the five time perspective dimensions, all paths from the time perspective dimensions to psychological well-being, positive affect, and negative affect, and their standardized parameter estimates. Chi-square = 3.85, df = 1, p = .05; comparative fit index = 1.00; incremental fit index = 1.00; normed fit index = 1.00; root mean square error of approximation = 0.08.

Since the results suggested that past positive and past negative time perspective contributed differently to psychological well-being, the two correlation coefficients (.41 and −.58) were Fisher-transformed and then compared with each other using a z-test (Steiger, 1980). This analysis was performed in order to investigate which of the correlations was stronger. The correlation between past negative and psychological well-being was found to be significantly higher than that for past positive and psychological well-being (z = 17.43, p < .001).

Discussion

The aim of the present study was to investigate the relationship between time perspective dimensions and temporal satisfaction with life, affect, and psychological well-being. As expected, satisfaction with past life was related to past perspective (i.e., Past Positive and Past Negative). This is understandable since individuals who report feeling pleasure, nostalgia, and happiness when recalling the past (i.e., Past Positive) should report feeling more satisfied with their past life. Likewise, rumination of bad moments from the past and generalization of bad past outcomes to the present and to the self (i.e., Past Negative) ought to predict lower satisfaction with the past. This is, for instance, related to research on explanatory style—individuals who internalize and interpret past negative events as long-lasting and associated to all aspects of their life are prone to depression (Buchanan & Seligman, 1995). Present life satisfaction was predicted by the tendency to take risks in order to achieve more positive emotions or avoid boredom in the present (i.e., Present Hedonistic). Consistent with previous suggestions (Kim-Prieto et al., 2005; Garcia & Erlandsson, 2011), present life satisfaction was predicted by how one feels about the past. Satisfaction with the future was, also as expected, predicted by the tendency to plan for the future, goal-setting, punctuality, and meeting deadlines (i.e., the Future time perspective dimension). As in previous studies (Garcia, Rosenberg & Siddiqui, 2011; Pavot, Diener & Suh, 1998), future life satisfaction was also related to individuals’ view of the past and the present. However, whereas the view of the past influenced satisfaction also in the present and future, the Present Hedonistic dimension did not influence the view of the past. This suggests that the Present Hedonistic dimension expresses a “here and now” approach to life.

In sum, the Past Negative time perspective dimension predicted lower levels of temporal life satisfaction as a whole, the Past Positive time perspective dimension predicted both past and future life satisfaction, and the Present Hedonistic time perspective dimension predicted both present and future life satisfaction (see Fig. 3). Our findings extend previous results concerning the relationships between time perspectives and subjective well-being (e.g., Boniwell et al., 2010) by specifying in which time frame life satisfaction is evaluated.

Figure 3 Summary of the results with regard to the relationships between the different time perspectives and temporal life satisfaction dimensions.

We also found that frequently feeling proud, interested, and engaged (i.e., positive affect) was associated with the tendency to take risks in order to achieve more positive emotions or avoid boredom in the present (i.e., the Present Hedonistic time perspective dimension), and approaching behavior related to the Future time perspective dimension. This is in line with suggestions (Sirois, 2014) with regard to the other end of this affect dimension and its relation to procrastination, namely, low positive affect’s role in the explanation of why procrastination is associated with less focus on the future. Positive affect, as measured here, is indeed labeled as high activation affect (Russell & Carroll, 1999). Accordingly, Schütz and colleagues (2013) showed that agentic behavior (i.e., goal-setting, planning for the future, self-control, etcetera) is associated with a self-fulfilling state defined as frequently experiencing high activation positive affect and infrequently experiencing high activation negative affect (see also Garcia, Anckarsäter & Lundström, 2013). In contrast, the Past Negative and the Present Fatalistic time perspective dimensions attenuated positive affect. In other words, having a negative view of the past (i.e., Past Negative) and feeling hopeless and lacking control were associated with experiencing few positive emotions. Moreover, a negative view of the past (i.e., Past Negative) and present (Present Fatalistic) predicted frequently feeling afraid, nervous and irritated, while a hedonistic present perspective (i.e., Present Hedonistic) attenuated these negative emotions.

Finally, psychological well-being was predicted by all time perspective dimensions. Memories of the past as being happy (i.e., Past Positive), taking risks in the present in order to achieve happy feelings and/or avoid boredom (i.e., Present Hedonistic), and a Future time perspective associated to agentic values (i.e., goal-setting, self-control, planning for the future, etcetera) were associated with higher levels of psychological well-being. In contrast, ruminating about negative events (i.e., Past Negative) and a hopeless and pessimistic view about the present (Present Fatalistic) was associated with low levels of psychological well-being. Interestingly, the effect of a Past Negative time perspective on psychological well-being was stronger than the effect of a Past Positive time perspective, thus supporting the view that the psychological effects of negative events are stronger than those of positive events (Baumeister, Catanese & Vohs, 2001). Along these lines, whereas a positive view of the past influenced past and future life satisfaction as well as psychological well-being, the effect of a negative view of the past influenced a larger number of variables—in fact all variables tested. Past Negative affected life satisfaction in all time frames, positive affect, negative affect and psychological well-being. This suggests that in order to improve general well-being, it may be more effective to reduce a Past Negative time perspective than by inducing a Past Positive time perspective. To summarize the results of the second model, Past Negative, Present Hedonistic, and Present Fatalistic time perspectives were related to affect and psychological well-being. The Future time perspective was related to both positive affect and psychological well-being, while the Past Positive time perspective only predicted psychological well-being (see Fig. 4).

Figure 4 Summary of the results with regard to the relationships between the different time perspectives, psychological well-being, and affect.

Limitations and suggestions for future research

The present study used self-reports which does not allow us to infer whether or not time perspective causally predicts actual well-being. Ryff’s multidimensional approach is, for instance, suggested to describe the fully functional individual (Ryff & Keyes, 1995). Thus, a balanced time perspective (i.e., the ability to hold past, present, and future time perspectives at the same time and to use them flexibly) may be caused by individuals’ tendency to accept all parts of her/his personality (i.e., high self-acceptance), see life as an opportunity to develop (i.e., personal growth), have a sense of meaning in life (i.e., purpose in life), have control in life (i.e., environmental mastery), have autonomy, and establish and keep positive relations with others. Another limitation is the fixed order in which the questionnaires were administered, after all, randomization of the order in which the instruments are presented to participants is suggested to ensure that responses to survey questions are not affected by the order of the instruments (Lavrakas, 2008). Also, we used the total psychological well-being score instead of each of the subscales because the short version used here has been found to have low reliability in previous studies (e.g., Clarke et al., 2001; Garcia & Siddiqui, 2009). A more reliable version may help disentangle which time perspective is associated with each psychological well-being dimension.

Concluding remarks

High levels of both subjective and psychological well-being were found to be related to memories of a happier and less sinister past, a more hedonistic and less pessimistic present, as well as to a more structured future.

“Happiness can be found, even in the darkest of times, if one only remembers to turn on the light”

Albus Dumbledore in Harry Potter and the Prisoner of Azkaban

We would like to thank Björn Mikmar and his most helpful staff at Friskis&Svettis Karlskrona/Ronneby for their collaboration in the collection of the data. We would also like to convey our gratitude to the participants for spending their valuable time answering the questionnaires, Erik Lindskär for his help with the data collection, and Nabeel Abd Algafoor at the University of Mustansiryah, Bagdad, for his statistical advice. Last but not the least we want to thank Sophia Isabella Garcia Rosenberg and Linnéa Mercedes Garcia Rosenberg for their help with Figs. 3 and 4.

Additional Information and Declarations

Competing Interests

Author Contributions

Ethics

Data Deposition

The authors declare that they have no competing interests.

Uta Sailer, Patricia Rosenberg, Amelie Gamble and Tommy Gärling wrote the paper, reviewed drafts of the paper.

Ali Al Nima performed the experiments, analyzed the data, contributed reagents/materials/analysis tools, wrote the paper, prepared figures and/or tables, reviewed drafts of the paper.

Trevor Archer conceived and designed the experiments, performed the experiments, wrote the paper, reviewed drafts of the paper.

Danilo Garcia conceived and designed the experiments, performed the experiments, analyzed the data, contributed reagents/materials/analysis tools, wrote the paper, prepared figures and/or tables, reviewed drafts of the paper.

The following information was supplied relating to ethical approvals (i.e., approving body and any reference numbers):

Data collection conformed to the Declaration of Helsinki and the Ethics Committee of the University of Gothenburg approved the research protocol. Verbal informed consent was obtained from all the study participants as agreed by the review board.

The following information was supplied regarding the deposition of related data:

Researchgate: https://www.researchgate.net/publication/260454569_Sailer_et_al._2014_Time_Perspective_and_Well-Being.

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
