# Peer review of "A happier and less sinister past, a more hedonistic and less fatalistic present and a more structured future: time perspective and well-being"

_PeerJ, doi:10.7717/peerj.303_

## Round 0.1 · original submission · Minor Revisions

The article is interesting but needs some minor revisions to answer the reviewers questions.

Reviewer 1 ·

Basic reporting

The paper is interesting because the aim is to study relationship between time perspective and psychological well-being. The topic is actual and new. I recommend to authors response to several questions.

Experimental design

Adequate. (see comments to author)

Validity of the findings

Good. (see comments to author)

Additional comments

Lines 22-32, authors indicate Time Perspective (TP) correlates with several behaviours , mood…. following a hypothetical biological mechanism to explain how past and future are related. I think that authors could include some of the more recent correlates of TP, from the first studies related to addiction (Fieulane, France), procrastination and morningness-eveningness (Diaz-Morales, Spain), environmental behavior (Milfont, ), etc.
Line 53. Please, indicate better the difference between Diener and Riff approach.
Line 64. Authors could related the “approach-related positive affect” with other personality models (e.g. Gray).
Line 102. Please explain why the order of application wasn’t counterbalanced. Early studies about mood and order of application could be refereed here. Include this question as limitation of study.
Line 119. Has been published other ZTPI versions with good psychometric properties? For example, in France, Spain, Italy, etc. Perhaps these another versions could be indicated here or in Introduction. Please, see page web of TP European research group.
Line 155. Typographical errors.
Lines 176-190. Authors test the fix of several models but the question is what is the model with better fix ? Please, calculate some statistic to contrast fix of models.
Lines 192-195. Both sentences are not supported by data analysis. Please, re-write or make the comparative analysis contrasting the structural models.
Lines 224. Explain why tow correlation coefficients are statistically compared.
Lines 236-239. Authors call to psychological variables to explain results. This might be related to Introduction, where these variables, process or models could be detailed.
Line 257. Please, include here some reference to the research realized by F. Sirois in relation to positive and negative affect and time perspective and procrastination.

·

Basic reporting

No comments.

Experimental design

No comments.

Validity of the findings

No comments.

Additional comments

No comments.

Reviewer 3 ·

Basic reporting

The article meets the basic standard but there are some small errors in the text see line 34, 367, 400, 402-4, 415 and 429. Except for that I have no other comments.

Experimental design

The question I ask is how the authors got in contact with respondents and how many of the contacts that were taken led to a participation in the study. So, what is the actual response rate?
But appart from that, no comments.

Validity of the findings

No Comments

Additional comments

The reading of Table 2 would be improved if the R2 values were added after one and each group of results (i. e. past life satisfaction, present life satisfaction and future life satisfaction in the Satisfaction Model; and PA, NA and PWB in the PANA and PWB model)

---

## Round 0.2 · accepted · Accept

This is a valuable contribution to the literature and shows the relevance of all aspects of a person's life (past, present, and future) for well-being.